# Voltage-controlled topological spin textures in the monolayer limit

Yangliu Wu[1,2,7], Bo Peng[1,2,7] ✉, Zhaozhuo Zeng[3,7], Chendi Yang[4], Haipeng Lu[1,2], Peiheng Zhou[1,2], Jianliang Xie[1,2], Difei Liang[1,2], Linbo Zhang[1,2], Peng Yan[3] ✉, Haizhong Guo[5,6] ✉, Renchao Che[4] ✉ & Longjiang Deng[1,2] ✉

The physics of phase transitions in low-dimensional systems has long been a subject of significant research interest. Long-range magnetic order in the strict two-dimensional limit, whose discovery circumvented the Mermin-Wagner theorem, has rapidly emerged as a research focus. However, the demonstration of a non-trivial topological spin textures in two-dimensional limit has remained elusive. Here, we demonstrate the out-of-plane electric field breaks inversion symmetry while simultaneously modulating the electronic band structure, enabling electrically tunable spin-orbit interaction for creation and manipulation of topological spin textures in monolayer CrI$_3$. The realization of ideal two-dimensional topological spin textures may offer not only an experimental testbed for probing the Berezinskii–Kosterlitz–Thouless mechanism, but also potential insights into unresolved quantum phenomena including superconductivity and superfluidity. Moreover, voltage-controlled spin-orbit interaction offers a novel pathway to engineer two-dimensional spin textures with tailored symmetries and topologies, while opening avenues for skyrmion-based next-generation information technologies.

The pursuit of low-dimensional (2D) magnetic materials and their unique topological spin textures dates back to the last century. Early theoretical studies recognized that while the Mermin-Wagner theorem prohibits long-range magnetic order in low-dimensional systems ($d < 3$) with continuous symmetry[1], this limitation could be overcome through substantial magnetic anisotropy. This is evidenced by recent realizations of Ising-like 2D magnets[2,3], such as monolayer ferromagnetic CrI$_3$ and Cr$_2$Ge$_2$Te$_6$, which have since sparked extensive research interest in this field. The Berezinskii–Kosterlitz–Thouless (BKT) theory predicts the stabilization of topological spin textures in rigorously two-

dimensional systems[4,5]. Through the breaking of inversion symmetry, the existence of skyrmions has been observed in van der Waals nanosheets via the topological magneto-optical effect[6,7]. However, conclusive experimental verification of topological spin textures in atomically thin monolayers remains a significant challenge. The topological spin textures in real-space and band structures of non-trivial topology in momentum space, have attracted enormous attention due to their elegant Berry curvature physics[8–16]. Topologically protected, particle-like spin textures—particularly skyrmions, which exhibit robust stability and controlled motion under electrical stimuli

[1]National Engineering Research Center of Electromagnetic Radiation Control Materials, School of Electronic Science and Engineering, University of Electronic Science and Technology of China, Chengdu, China. [2]Key Laboratory of Multi Spectral Absorbing Materials and Structures of Ministry of Education, School of Electronic Science and Engineering, University of Electronic Science and Technology of China, Chengdu, China. [3]School of Physics and State Key Laboratory of Electronic Thin Films and Integrated Devices, University of Electronic Science and Technology of China, Chengdu, China. [4]Laboratory of Advanced Materials, Department of Materials Science, Collaborative Innovation Center of Chemistry for Energy Materials(iChEM), Fudan University, Shanghai, China. [5]School of Physics, Zhengzhou University, Zhengzhou, P. R. China. [6]Institute of Quantum Materials and Physics, Henan Academy of Sciences, Zhengzhou, China. [7]These authors contributed equally: Yangliu Wu, Bo Peng, Zhaozhuo Zeng. ✉e-mail: bo_peng@uestc.edu.cn; yan@uestc.edu.cn; hguo@zzu.edu.cn; rcche@fudan.edu.cn; denglj@uestc.edu.cn

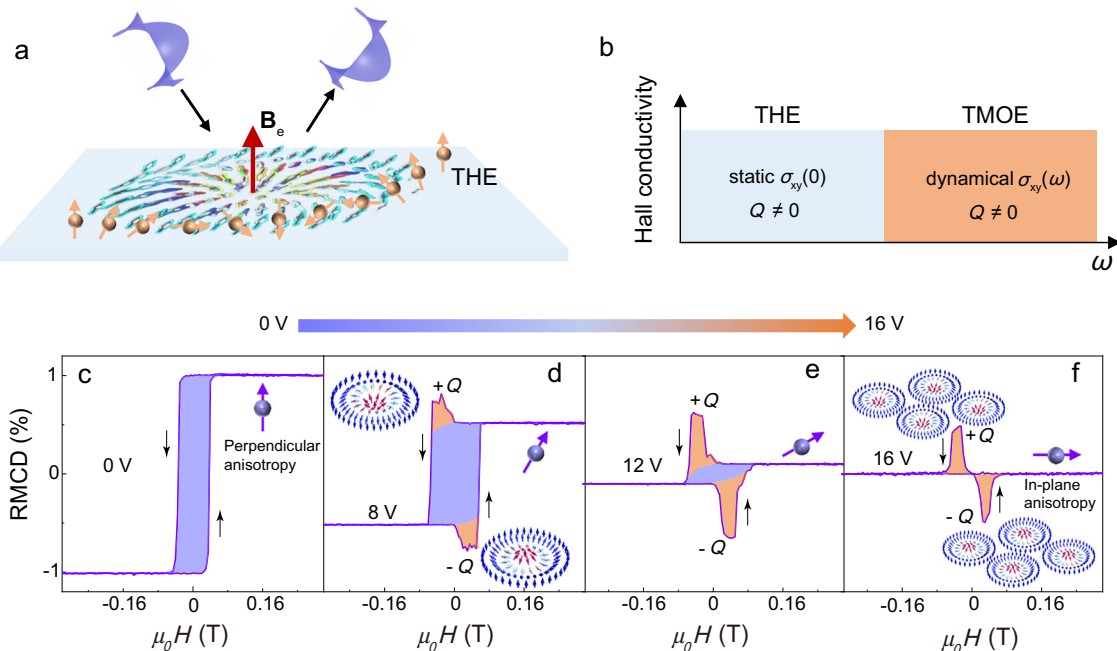

**Fig. 1 | Voltage-controlled topological magnetic circular dichroism in monolayer CrI₃. a** Sketch of the topological Hall effect and topological magnetic circular dichroism induced by the emergent magnetic field. **b** Summary of the two topological Hall regimes. **c–f** RMCD versus magnetic field at different gate voltages at 10 K. The black arrows indicate the direction of the field sweep, while the purple arrows with spheres schematically represent the magnetic anisotropy.

(e.g., applied currents)—represent promising candidates for information carriers in next-generation memory and logic devices[17–20]. Among the key challenges in developing future energy-efficient devices is achieving magnetic state switching via electric fields[21,22]. Despite early demonstrations of electric-field switching of skyrmions through STM tips[23,24], reports on their translation into practical tunnel junction nanodevices remain scarce[25].

Here, we demonstrate the electrical creation of the topological spin textures in monolayer CrI₃, which we detect through the unconventional topological magnetic circular dichroism. The gate-controlled modulation of both Dzyaloshinskii-Moriya interaction (DMI) and magnetic anisotropy energy (MAE) drives a reversible phase transition from ferromagnetic order to stabilized topological spin states. Electrostatic gating enables continuous modulation of the MAE from perpendicular to in-plane configurations, with the sign reversal of MAE permitting voltage-controlled stabilization of topological spin textures across a well-defined biasing regime. Our results provide a platform for understanding topological magnetic quasiparticles and their interactions in two-dimensional systems. This may contribute to elucidating the mechanisms of the BKT transition and related fundamental quantum phenomena such as superfluidity and superconductivity[26,27]. Besides, the electric-field-controlled creation and annihilation of skyrmions break through the limitations of the energy-intensive current-driven paradigm in racetrack memory[23–25], which may revolutionize next-generation memory devices and information technologies.

## Results and discussion

The scalar spin chirality of topological magnetic structures has manifested itself as an emergent magnetic field ($\mathbf{B}_e$)[14], which gives rise to a transversal electrical response (Fig. 1a) known as the conventional topological Hall effect (THE) at zero-frequency limits[28,29]. The emergent magnetic field arising from skyrmion potentially gives rise to the topological magneto-optical effect (TMOE)[30,31], which can be considered as the optical-frequency counterpart of the conventional THE (Fig. 1b). Tunable optical-frequency permit the

identification of topological magnetic structures in diverse material systems—including metals, semiconductors and insulators—thereby extending the utility of the topological magneto-optical effect beyond the metallic regimes accessible to conventional THE. To investigate magnetoelectric coupling in the ideal 2D limit, we engineered monolayer CrI₃ devices supporting both perpendicular electric fields and electrostatic doping. Monolayer CrI₃ was exfoliated mechanically from bulk crystals onto a hBN flake on 285-nm-thick SiO₂/Si substrates in a nitrogen-filled glove box (see "Methods"). A micrograph of the monolayer CrI₃ and the device is shown in Supplementary Fig. 1a, b. Supplementary Fig. 1c illustrates the structure of a representative monolayer CrI₃ device. This vertically stacked configuration consists of a monolayer of CrI₃, hBN flakes and graphene contacts, in which the hBN flakes act as the dielectric layers for providing electrostatic doping and electric field. The dielectric layer of hBN has a thickness of 9.8 nm (Supplementary Fig. 2) and a dielectric constant of 2.9 (Supplementary Fig. 3; see "Methods"), consistent with the previous reports[32,33]. Reflective magnetic circular dichroism (RMCD) measurements were performed on a designated region of the device, using a laser with a wavelength of 633 nm and a fixed power of 3 μW (see "Methods"). Figure 1c shows a representative RMCD signal of the monolayer CrI₃ device as a function of the out-of-plane magnetic field ($\mu_0 H$) at zero gate voltage, with black arrows indicating the direction of the magnetic field sweep. The results of monolayer CrI₃ are in agreement with the previous reports[1]. Namely, monolayer CrI₃ exhibits ferromagnetism with a coercive field ($\mu_0 H_c$) of approximately 0.04 T and a Curie temperature ($T_c$) close to 40 K (Supplementary Fig. 4). And the absence of a high-field step in the RMCD loop (Supplementary Fig. 5a) excludes interlayer antiferromagnetic coupling and validates the monolayer[34]. Furthermore, the polarization-resolved and temperature-dependent Raman measurements of spin-lattice coupling further confirm the monolayer feature[35,36], in which the $A_{1g}$ phonons are only detected at 127.8 cm⁻¹ in cross- (XY) and co-linear (XX) polarization configurations and significantly increased in XY configuration below 40 K. This temperature-dependent evolution of the polarization-resolved

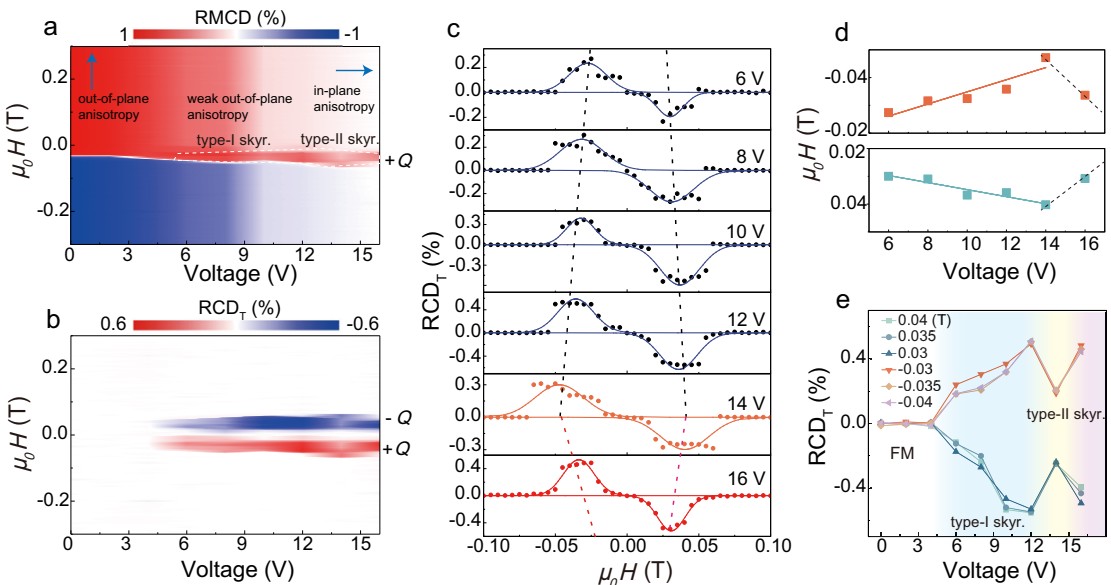

**Fig. 2 | Voltage-controlled multistage topological magnetic phase transition in monolayer CrI$_3$. a** Intensity of the RMCD signal of a monolayer CrI$_3$ device as a function of both gate voltage and applied magnetic field (sweeping from positive to negative). The blue arrows highlight the voltage-induced evolution of the perpendicular magnetic anisotropy to the in-plane anisotropy. **b** topological magnetic phase diagram from the RCD$_T$ as a function of gate voltage $V_g$ and external field $\mu_0H$. **c** The RCD$_T$ loops measured at various voltages, selected from Fig. 2b. Solid dots represent the experimental values, while the solid line is derived from Gaussian fitting. **d** The RCD$_T$ peak positions, where the density of topological

quasiparticles is maximal under the corresponding magnetic fields, are plotted as a function of the applied voltage. The solid lines represent linear fits, while the dashed lines serve as visual guides. **e** Under selected magnetic fields of $\mu_0H = \pm 0.04$ T, $\pm0.035$ T, and $\pm0.03$ T, the relationship between RCD$_T$ and voltage indicates a voltage-induced multistage magnetic phase transition from the ferromagnetic phase to the skyrmion phase. Type-I skyrmions are defined as those stabilized by an out-of-plane magnetic field in systems with perpendicular magnetic anisotropy, whereas Type-II skyrmions are formed under in-plane magnetic anisotropy with the assistance of an out-of-plane magnetic field.

Raman features and the RMCD signals provides explicit evidence that the CrI$_3$ sample is a monolayer (Supplementary Note 1, Supplementary Figs. 4, 5).

Figure 1d shows the RMCD signal as a function of the out-of-plane magnetic field under an applied voltage of 8 V, revealing a pair of anomalous RMCD peaks near the spin-switching transition field. The polar RMCD signal intensity ($I_{RMCD}$) is approximately proportional to $\sigma_{xy}(\omega)$, as expressed by the following equation[37–39]

$$I_{RMCD} = \frac{2Z_0 d\sigma_{xy}(\omega)}{1 - (n_S + Z_0 d\sigma_{xx}(\omega))^2} \qquad (1)$$

Here, $Z_0$ is the impedance of free space, $n_s$ denotes the refractive index of the substrate, and $d$ represents the thickness of the monolayer CrI$_3$, approximately 0.7 nm; $\sigma_{xx}(\omega)$ and $\sigma_{xy}(\omega)$ are the longitudinal and transverse components of the optical Hall conductivity, respectively. Analogous to the analytical framework for the conventional topological Hall effect, $\sigma_{xy}(\omega)$ can be separated into two primary components: one originating from ferromagnetism, denoted as $\sigma_{xy}^A(\omega)$, and the other arising from topological spin textures, denoted as $\sigma_{xy}^T(\omega)$. The $\sigma_{xy}^A(\omega)$ is proportional to the magnetization (**M**), while $\sigma_{xy}^T(\omega)$ is proportional to the emergent magnetic field **B**$_e$ induced by topological spin textures[31], which can be determined by evaluating the solid angle traced by the spin moment **n** as it winds within the $xy$-plane[15], given by

$$\sigma_{xy}^T(\omega) \propto \mathbf{B}_e = \frac{1}{2}\mathbf{n} \cdot \left(\frac{\partial \mathbf{n}}{\partial x} \times \frac{\partial \mathbf{n}}{\partial y}\right)\hat{\mathbf{z}} \qquad (2)$$

For a skyrmion, **B**$_e$ is quantized: the integral over the 2D space is simply $2\pi$ times the topological charge ($Q$) of skyrmions, $Q = \frac{1}{4\pi}\int \mathbf{n} \cdot \left(\frac{\partial \mathbf{n}}{\partial x} \times \frac{\partial \mathbf{n}}{\partial y}\right)dxdy$. Therefore, spin textures with non-zero $Q$ can induce differences in the reflectivity between the left- and right-

handed circularly polarized lights, even in the absence of a net magnetic moment (Fig. 1a). We refer to this effect as reflective topological circular dichroism (RCD$_T$), which can also lead to the rotation of the polarization direction of linearly polarized reflected light, known as the topological Kerr effect. The RMCD loops illustrated in Fig. 1e can be expressed as the sum of two components,

$$I_{RMCD} = I_{RCD(M)} + I_{RCD(T)} \qquad (3)$$

Here, $I_{RCD(M)}$ is the intensity of conventional reflective circular dichroism (RCD$_M$), which serves as a measure of the out-of-plane net magnetization of the ferromagnetic background (highlighted in purple, see Fig. 1c–f), while $I_{RCD(T)}$ is the intensity of RCD$_T$ and is correlated with the density of $Q$ (highlighted in orange, Fig. 1d–f). The upward and downward RMCD peaks suggest the generation of the topologically nontrivial skyrmions with $+Q$ and $-Q$, respectively[15,16], in the vicinity of the spin-switching field[40]. Figure 1e illustrates the relationship between the magnetic field and the RMCD signal at an applied voltage of 12 V. The residual signal of RCD$_M$ hysteresis is significantly reduced, and the saturation magnetic field is increased, suggesting a decrease in the perpendicular magnetic anisotropy (PMA)[41]. Meanwhile, RCD$_T$ becomes more pronounced due to the increased density of $Q$. Remarkably, under a voltage of 16 V, the RCD$_M$ hysteresis completely vanishes, leaving only a significant RCD$_T$, accompanied by a giant topological circular dichroism (Fig. 1f), akin to the giant topological Hall effect observed in skyrmion lattices[15,16]. This indicates that the perpendicular anisotropy is lost, and due to electron doping[41,42], the system transitions to a small in-plane anisotropy, while the density of skyrmions significantly increases with the assistance of an out-of-plane magnetic field. Similar voltage-controlled RMCD phenomena were also observed in another monolayer CrI$_3$ device (Supplementary Fig. 6).

Figure 2a presents the RMCD intensity as a function of gate voltage ($V_g$) and magnetic field ($\mu_0H$), sweeping from positive to negative

values. In this magnetic phase diagram, white regions indicate near-zero RMCD intensity, corresponding to an in-plane ferromagnetic state. Darker red and blue areas represent two out-of-plane ferromagnetic states with strong RMCD signals of opposite signs. Light blue and red regions denote canted ferromagnetic states with opposite out-of-plane components. Notably, a distinct red stripe appears near the critical magnetic field for the spin-up to spin-down transition, identified as a topological magnetic phase with positive topological charge. In the non-topological phase region, increasing voltage drives a smooth transition from out-of-plane to in-plane ferromagnetism, suggesting a voltage-induced evolution of the perpendicular magnetic anisotropy toward in-plane anisotropy (highlighted by the blue arrows)[42]. The corresponding phase diagram, obtained by sweeping the magnetic field from negative to positive values, is shown in Supplementary Fig. 7a. Similar to Fig. 2a, a distinct blue stripe appears near the critical magnetic field for the spin-down to spin-up transition, representing a topological magnetic phase with negative topological charge. Figure 2b displays the intensities of the topological circular dichroism $RCD_T$ as a function of $V_g$ and $\mu_0 H$, revealing a critical voltage for the emergence of the topological magnetic orders near 5 V. Additionally, the critical magnetic field distinguishing the ferromagnetic and topological magnetic phases varies with the applied voltage. $RCD_T$ is obtained by subtracting $RCD_M$ from the total RMCD, as described in Eq. (3) and Fig. 1e. Figure 2c shows the $RCD_T$ loops measured at several representative voltages, extracted from Fig. 2b. As the voltage increases, the $RCD_T$ peaks with positive and negative topological charges shift linearly toward higher magnetic fields, with slopes of 22 Oe $V^{-1}$ and 13 Oe $V^{-1}$, respectively (Fig. 2d). This linear shift demonstrates precise and continuous voltage-controlled tuning of the topological magnetic phases. This behavior arises from the reduction of the perpendicular anisotropy induced by higher voltages, thus necessitating a stronger out-of-plane magnetic field to stabilize the skyrmions. However, above 14 V, the positions of the $RCD_T$ peaks abruptly shift toward lower magnetic fields. It suggests a transition from the perpendicular magnetic anisotropy to in-plane one, thereby causing a sudden shift in the magnetic field required to stabilize the topological magnetic structure. Figure 2e illustrates the electrically controlled $RCD_T$ under three selected magnetic fields with opposite directions, extracted from Fig. 2b. The $RCD_T$ signal remains near zero at voltages below approximately 5 V, corresponding to the ferromagnetic ordering. As the voltage increases to a critical value around 5 V, the $RCD_T$ signal rises sharply, indicating an electrical switch from trivial ferromagnetic to topological magnetic states. With voltage further increasing, the $RCD_T$ signal gradually rises and saturates, suggesting an increase in skyrmion density. Near 14 V, the $RCD_T$ signal undergoes a sudden change, suggesting that a fundamental shift in the magnetic anisotropy may trigger the nucleation of the skyrmions with new characteristics, thereby causing a sharp variation in the $RCD_T$ signal. We refer to the skyrmions induced by an out-of-plane magnetic field under perpendicular anisotropy as Type-I skyrmions, and those formed under a small in-plane magnetic anisotropy induced by an out-of-plane magnetic field as Type-II skyrmions (Figs. 2a and 2e). This abrupt behavior at 14 V is consistent with the voltage-induced shifts observed in $RCD_T$ peak positions (Fig. 2d) and intensities (Supplementary Fig. 7b). Furthermore, the increasing voltage results in a corresponding rise in $RCD_T$, attributed to an increase in the density of type-II skyrmions. The voltage can continuously regulate the density and morphology of the topological magnetic quasiparticles, and one of the key reasons is that the voltage enables continuous adjustment of the magnetic anisotropy energy, even changing its sign[42,43], leading to a gradual transition from perpendicular anisotropy to in-plane anisotropy (Supplementary Fig. 8c).

The stability of the skyrmions is governed by two critical parameters: the DMI and magnetic anisotropy, both originating from the fundamental mechanism of the spin–orbit coupling (SOC)[43–45].

Previous theoretical predictions suggested that skyrmions could be generated in a monolayer magnet through either out-of-plane electric fields or geometric defect-induced inversion symmetry breaking[46,47]. However, the exclusive consideration of DMI without incorporating magnetic anisotropy modulation may critically hinder skyrmion stabilization. This fundamental constraint most likely underlies the absence of experimentally observed skyrmions in the monolayer limit. As illustrated in Supplementary Fig. 9a, the application of an out-of-plane electric field in monolayer $CrI_3$ breaks inversion symmetry, leading to the emergence of a significant DMI[48]. The magnetic anisotropy energy (MAE) in $CrI_3$ monolayers is intimately associated with the spin and orbital structures near the Fermi level. First-principles calculations have demonstrated that the MAE is primarily governed by the indirect SOC, with the direct SOC contribution being negligible[49]. By the perturbation theory, the MAE can be succinctly expressed as[50]

$$\text{MAE} = \xi^2 \sum_{\mu, o, \sigma, \sigma'} \sigma\sigma' \frac{|\langle o, \sigma|L_z^l|\mu, \sigma'\rangle|^2 - |\langle o, \sigma|L_x^l|\mu, \sigma'\rangle|^2}{E_{\mu,\sigma} - E_{o,\sigma'}} \tag{4}$$

Here, $u$ and $o$ denote the unoccupied and occupied states, respectively, $E_{u/o,\sigma}$ represents the band energy of these states, and the spin indices $\sigma/\sigma'$ range over ±1, corresponding to the two orthogonal spin states at the $k$-point. Previous theoretical investigations have established that $|\langle o, \sigma|L_z^l|\mu, \sigma'\rangle|^2 - |\langle o, \sigma|L_x^l|\mu, \sigma'\rangle|^2$ is negative[51]. In addition to the contribution from the orbital angular momentum, the relative spin polarization ($\sigma, \sigma'$) of the states near the Fermi level must also be considered. When $CrI_3$ is slightly n-doped, the two lowest conduction bands act as the unoccupied and occupied states, with their parallel spin alignment ($\sigma\sigma' = 1$, right panel in Supplementary Fig. 9b) resulting in a negative MAE, which favors an in-plane magnetic anisotropy. In stark contrast, under slight p-doping, the dominant contributions originate from the two highest valence bands with oppositely aligned spins ($\sigma\sigma' = -1$, left panel in Supplementary Fig. 9b), resulting in a positive MAE that favors perpendicular magnetic anisotropy. In our experimental observations, the application of a positive voltage results in electron doping and further induces a transition from perpendicular magnetic anisotropy to in-plane anisotropy in monolayer $CrI_3$, while a negative voltage (hole doping) strengthens the perpendicular out-of-plane magnetization (Supplementary Fig. 8). Theoretically, MAE ($K$) must satisfy $K \le \pi^2 \cdot D^2/16 \cdot J$ for enabling the formation of a skyrmion lattice[43–45], implying that the skyrmions are stabilized only within a regime of weak anisotropy. Here, $D$ denotes the Dzyaloshinskii–Moriya interaction energy, while $J$ represents the exchange interaction energy. Supplementary Fig. 9c illustrates the relationship between the factor $\eta = \pi^2 \cdot D^2/16 \cdot J$, MAE ($K$), and gate voltage, where the $D$ term is induced by the electric field and the $K$ term arises from electrostatic doping[42,48]. Using a simple parallel-plate capacitor model[52,53], the corresponding voltage values can be inferred from the electric field strength and doping concentration within the $CrI_3$ monolayer (see "Methods"). This approach enables a comparative analysis of the voltage ranges for skyrmion stabilization between theoretical predictions and experimental observations. The theoretically predicted voltage thresholds for type-I ($0 < K < \eta$) and type-II ($K < 0$) skyrmion formation closely match the experimental voltages for both skyrmion types (highlighted in light yellow and purple areas).

The evolution of the magnetic structures in monolayer $CrI_3$ under out-of-plane magnetic fields at different voltages was simulated by solving the Landau-Lifshitz-Gilbert (LLG) equation (see "Methods"). The parameters for atomic-scale spin dynamics simulations were selected based on the criteria for the skyrmion formation as illustrated in Supplementary Fig. 9c. Figure 3a presents the atomic-scale spin dynamics simulation results at a voltage of 16 V (in-plane anisotropy, $K < 0$), depicting the evolution of the magnetic domains as the out-of-plane magnetic field $B/J$ is swept from +0.214 to −0.214. As the magnetic field decreases from positive saturation to zero, the system

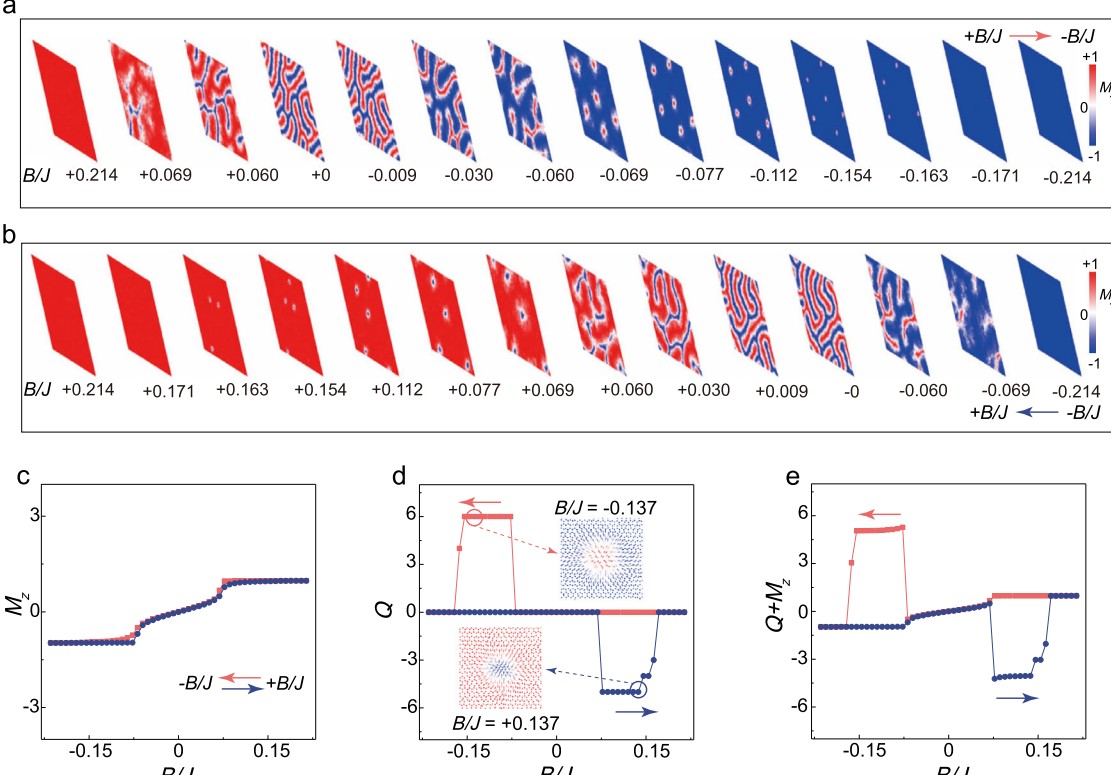

**Fig. 3 | Simulated evolution of the magnetic structures in monolayer CrI₃ induced by a magnetic field under a voltage of 16 V (in-plane anisotropy, $K < 0$). a** A series of representative magnetic domain configurations during the process of the magnetic field sweeping from the positive maximum to the negative maximum. A $100 \times 100$ supercell was used for the magnetic simulation of the two-dimensional spin lattice, with colors mapping the out-of-plane magnetic moment components ($M_z$). **b** A series of representative magnetic domain configurations, as the magnetic field sweeps back from the negative maximum to the positive maximum, is consistent with the RMCD loop sweeping. **c** The variation of $M_z$ with the magnetic field ($M_z$ normalized to its saturation value). The orange line represents the magnetic field sweeping direction from positive to negative, while the blue line indicates the opposite sweeping direction. **d** The evolution of the topological charge (per 10,000 spin sites) along the same hysteresis loop depicted in (**c**). The inset shows an enlarged view of the spin configuration of a single skyrmion, with arrows indicating the direction of the spins and colors representing $M_z$. **e** The sum of the topological charge and $M_z$ changes along the same hysteresis loop in (**c**, **d**), which resembles the RMCD loop observed experimentally under a voltage of 16 V.

evolves from an up-magnetized single-domain state to stripe domains. With increasing negative field, these domains fragment into skyrmions with positive topological charge ($Q > 0$), which subsequently annihilate at higher negative fields, establishing a down-magnetized state. The reverse sweep ($-0.214 \rightarrow +0.214$, Fig. 3b) shows analogous but inverted behavior, with stripe domains forming negative-charge skyrmions ($Q < 0$) that disappear upon reaching positive saturation. As shown in Fig. 3c, the $M_z$ versus magnetic field curve shows no hysteresis, a characteristic resulting from the in-plane anisotropy ($K < 0$) induced by electron doping[42]. Figure 3d shows the topological charge density versus magnetic field. During positive-to-negative magnetic field sweeps, a positive topological charge peak emerges at the negative critical field. Conversely, negative-to-positive sweeps generate a negative peak at. This antisymmetric behavior correlates well with the field-dependent $RCD_T$ shown in Fig. 1f. Analysis of the magnetic structure in Fig. 3d inset reveals a Néel-type skyrmion configuration at fields near the topological charge peak. Figure 3e schematically demonstrates the variation of the sum of $M_z$ and the topological charge with the out-of-plane magnetic field, which aligns with the experimentally observed RMCD loop featuring anomalous peaks and no residual signal of $RCD_M$ (Fig. 1f). This confirms the core conclusion: the total RMCD signal intensity $I_{RMCD}$ can be expressed as Eq. (3) contributing to the net out-of-plane magnetization $M_z$, and the density of the topological charge $Q$. Supplementary Fig. 10a, b illustrate the evolution of the magnetic structure under an out-of-plane magnetic field at a voltage of 8 V (perpendicular

magnetic anisotropy, $K > 0$). The evolution of the magnetic structure resembles that observed at 16 V. However, due to the perpendicular anisotropy, the $M_z$ hysteresis loop shows a distinct opening (Supplementary Fig. 10c), indicating finite remanent magnetization. This observation is consistent with the residual RMCD signal detected at approximately 8 V (Fig. 1d). In contrast to the in-plane anisotropy at 16 V, the perpendicular anisotropy at 8 V yields lower skyrmion density, indicated by a weaker topological charge peak (Fig. 3d, Supplementary Fig. 10d), consistent with the voltage-dependent $RCD_T$ peak evolution (Fig. 2c, Supplementary Fig. 7b).

To further explore the properties of the voltage-induced two-dimensional topological magnetic phase, Fig. 4a displays the RMCD intensities as a function of the magnetic field at various temperatures under a 16 V voltage. As the temperature increases, the topological magnetic phase undergoes a transition to an in-plane ferromagnetic state at approximately 24 K, followed by a transition from the in-plane ferromagnetism to a paramagnetic state around 27 K. The Curie temperature ($T_c$) of monolayer CrI₃ under a 16 V voltage is lower than that of intrinsic monolayer CrI₃ (Supplementary Fig. 4) because the electron doping reduces the MAE (Fig. 3c). On the contrary, applying a voltage of −14V slightly increases the $T_c$ of CrI₃ (Supplementary Fig. 11) due to the enhancement of MAE through the hole doping[42]. This behavior is attributed to the fact that the $T_c$ of monolayer CrI₃ is primarily governed by magnetic anisotropy induced by the SOC effects[51]. Figure 4b, c shows the phase diagrams of topological magnetic orders in monolayer CrI₃ as functions of temperature and magnetic field. The

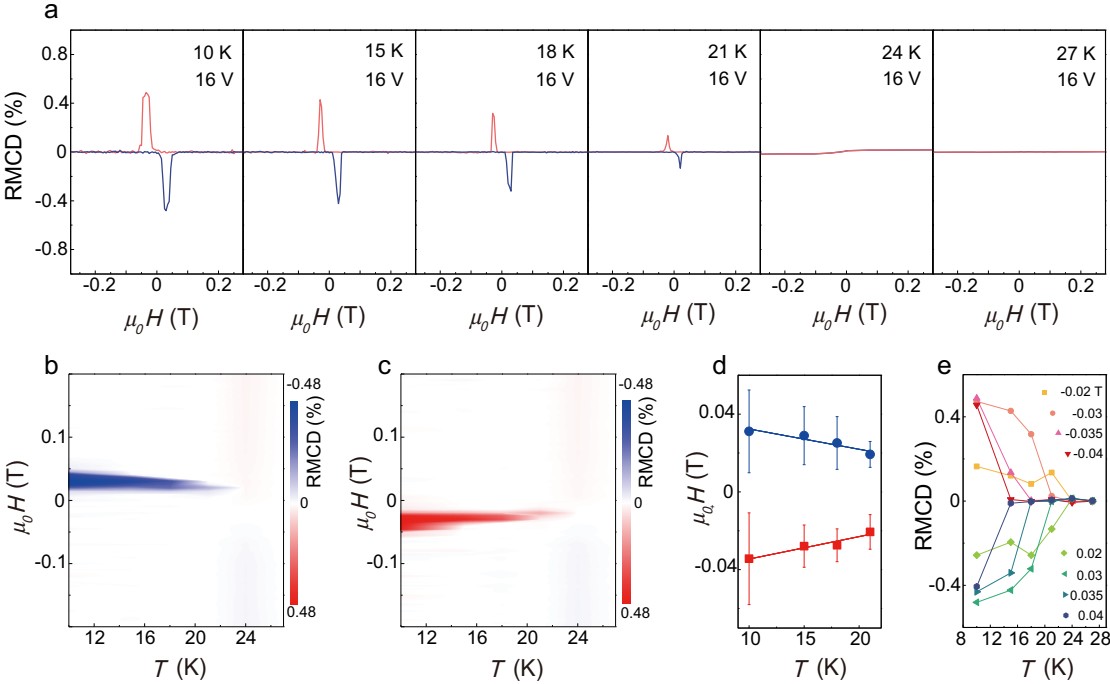

**Fig. 4 | Temperature-magnetic field phase diagram at 16 V. a** Intensity of the RMCD signal of a monolayer CrI$_3$ device under a 16 V voltage as a function of the magnetic field at different temperatures. The orange line represents the sweeping direction from positive to negative magnetic field, while the blue line represents the sweeping direction from negative to positive magnetic field. **b** The RMCD intensity as a function of temperature and magnetic field (sweeping from negative to positive). **c** The time-reversal process corresponding to (**b**) (sweeping from positive to negative). **d** The RCMD peak positions, where the density of topological quasi-particles is maximal under the corresponding magnetic fields, are plotted as a

function of the temperatures. Blue solid circles and red solid squares represent the magnetic field values corresponding to the blue and red solid RMCD peaks in (**a**), respectively. Solid lines denote linear fits to the experimental data, and error bars represent the half-peak widths. The peak positions and half-peak widths are obtained through Gaussian fitting. **e** The RMCD signal intensity as a function of temperature at various positive and negative magnetic fields indicates that the out-of-plane magnetic field significantly influences the phase transition temperature from topological magnetic to ferromagnetic.

results indicate that the magnetic field range for stabilizing the magnetic skyrmions decreases with increasing temperature, consistent with atomic-scale spin dynamics simulations (Supplementary Fig. 12, Supplementary Notes 2). Additionally, the magnetic fields corresponding to the maximum density of the topological quasiparticles— the positions of the RMCD peaks with positive topological charge (upward peaks) and negative topological charge (downward peaks)— decrease linearly with temperature, exhibiting slopes of 11.5 Oe K$^{-1}$ and 10.4 Oe K$^{-1}$, respectively (Fig. 4d). Furthermore, the half-peak width of the RMCD peaks, which characterizes the robustness of topological quasiparticles against magnetic fields, also gradually diminishes as the temperature rises. This behavior closely resembles the field-temperature phase diagrams of the skyrmion lattices in bulk Gd$_2$PdSi$_3$ and GdRu$_2$Ge$_3$[15,16]. Figure 4e illustrates the critical temperature behavior of the topological phase transition by comparing the temperature dependence of RMCD under different magnetic fields (horizontal cuts of the phase diagrams in Fig. 4b, c). The critical temperature shifts from approximately 15 K at ±0.04 T to 24 K at ±0.02 T. This indicates that as the out-of-plane magnetic field increases, the critical temperature decreases. The evolution of the topological magnetic phase induced by a 14 V voltage, as a function of temperature and magnetic field, is similar to that observed at 16 V (Supplementary Figs. 13, 14). However, compared to the 16 V case, a slight perpendicular magnetic anisotropy is persisted. Therefore, in addition to the significant RCD$_T$ peaks, an open RCD$_M$ loop was also observed (detailed discussion is provided in Supplementary Notes 3).

In summary, we demonstrated a voltage-control of the topologically protected spin textures in monolayer CrI$_3$. By utilizing the voltage-induced synergistic effects between DMI and MAE, we achieved the generation and continuous manipulation of the skyrmion

at the two-dimensional limit, providing a powerful tool for controlling the topological magnetic states at a single atomic layer limit. The results highlight the potential of the voltage-driven modulation of the spin-orbit interactions in creating a diverse range of two-dimensional magnetic textures with distinct topological properties. The precise electric-field control in engineering topological magnetic quasi-particles opens up new possibilities for ultra-high-density, ultra-low-power, and high-speed spintronic applications, while also broadening the scope of topological magneto-optical effects, thereby advancing the development of cutting-edge magneto-optical information storage and processing technologies.

## Methods

### Sample fabrication
Monolayer CrI$_3$ was mechanically exfoliated from bulk crystals using PDMS films in an inert glovebox. The bulk CrI$_3$ crystals were synthesized via chemical vapor transport from elemental precursors with a Cr molar ratio of 1:3. The exfoliated flakes of hBN, CrI$_3$ and graphene were sequentially transferred onto pre-patterned Au electrodes on SiO$_2$/Si substrates to form heterostructures. These samples were then in-situ loaded into an optical cryostat within the glovebox for magneto-optical-electric joint measurements. Throughout the fabrication and measurement processes, the CrI$_3$ samples were kept isolated from the atmosphere.

### Magneto-optical measurement
Polar RMCD and Raman measurements were conducted using a magneto-optical-electric joint-measurement scanning imaging system (MOEJSI)[54]. This system is based on a Witec Alpha 300 R Plus low-wavenumber confocal Raman microscope, integrated with a closed-

cycle superconducting magnet (7 T) and a cryogen-free optical cryo-stat (10 K), which features a specially designed sample mount and electronic transport measurement setup.

Raman signals were recorded using the Witec Alpha 300 R Plus, coupled with a closed-cycle He optical cryostat and a superconducting magnet. A 50× objective (NA = 0.45) was employed for measurements at 10 K and under magnetic field conditions. The signals were collected via a photonic crystal fiber and directed into the spectrometer with an 1800 g mm$^{-1}$ grating. Polarization-resolved Raman spectra were obtained by rotating the analyzer placed before the fiber. The excitation laser at 633 nm (1.96 eV) was set to an intensity of approximately 0.3 mW, with a typical integration time of 120 s.

For polar RMCD measurements, a 633 nm laser (~3 μW) modulated by a photoelastic modulator (PEM, 50 kHz) was reflected by a non-polarizing beamsplitter (R/T = 30/70) and focused onto the sample using a 50× objective (NA = 0.55, Zeiss). The reflected beam, collected by the same objective, passed through the beamsplitter and was detected by a photomultiplier tube (PMT), coupled to a lock-in amplifier, the Witec scanning imaging system, a superconducting magnet, and a voltage source meter.

### Estimation of the electric field strength and doping concentration

We evaluate the doping level and electric field strength of monolayer $CrI_3$ in our devices under electrostatic gating using a parallel-plate capacitor model. A simple heterostructure has been employed in this study: graphene/hBN/$CrI_3$/graphene (Fig. 1c). The doping density of $CrI_3$ ($n_{Cr}$) and the electric field ($E_{Cr}$) of $CrI_3$ are obtained under the applied voltage ($V$) between the two graphene electrodes.

$$E_{Cr} \approx \frac{V}{d_{Cr} + \frac{\varepsilon_{Cr} \cdot d_{BN}}{\varepsilon_{BN}}} \tag{5}$$

$$n_{Cr} \approx \frac{V}{4\pi k \left( \frac{d_{Cr}}{\varepsilon_{Cr}} + \frac{d_{BN}}{\varepsilon_{BN}} \right)} \tag{6}$$

Here, $d_{Cr}$ and $d_{BN}$ represent the thicknesses of $CrI_3$ and hBN, respectively, $\varepsilon_{Cr}$ and $\varepsilon_{BN}$ are the dielectric constants of $CrI_3$ and hBN, and $k$ is the electrostatic constant. The dielectric constant of hBN is measured to be 2.9 using a ferroelectric analyzer[54], while the dielectric constant of $CrI_3$ is 7[55]. The thickness of hBN is determined to be 9.8 nm using optical contrast methods, while the thickness of a monolayer of $CrI_3$ is 0.7 nm. By calculating Eqs. (4, 5), along with the theoretical predictions of the electric field–DMI function[48] and the doping concentration–magnetic anisotropy function[42], we are able to compare the experimentally observed voltage window for the emergence of topological magnetic order with the theoretically predicted critical voltage for its onset (Supplementary Fig. 9c).

### Optical contrast measurement

White light is normally incident on the hBN/SiO$_2$/Si structure and focused into a small spot using a 50×, NA = 0.45 microscope objective. The reflected light is then collected, passes through a beamsplitter, and is directed onto the entrance slit of a spectrometer. The spectrally resolved reflection signal is captured by a CCD at the spectrometer's exit aperture. In our experiments, we measure the optical contrast of the three-layer structure, which is defined as:

$$C = \frac{R_{SiO2} - R_{SiO2+hBN}}{R_{SiO2}} \tag{7}$$

Where $R_{SiO_2}$ represents the reflection coefficient at normal incidence for a bare SiO$_2$/Si substrate, while $R_{SiO_2+hBN}$ corresponds to the reflection coefficient for a substrate covered with hBN. This defines the normalized change in reflectivity of the hBN layer relative to the underlying substrate[56].

### Atomic-scale spin dynamics simulations

The numerical simulation were performed by solving the atomic-scale LLG equation $-\frac{1+\alpha^2}{\gamma}\frac{d\mathbf{m}_i}{dt} = \mathbf{m}_i \times \mathbf{H}_i + \alpha \mathbf{m}_i \times (\mathbf{m}_i \times \mathbf{H}_i)$ with periodic boundary condition, where the $\mathbf{m}_i$ is the reduced magnetization of the $i$-th Cr atom, $\gamma = 1.76 \times 10^{11}$ T$^{-1}$ s$^{-1}$ is the gyromagnetic ratio, $\alpha$ is the Gilbert damping constant (set to 1.0 to significantly reduce the evolution time) and $\mathbf{H}_i = -\frac{\delta \mathcal{H}}{\mu_s \delta \mathbf{m}_i}$ is the effective field with the magnetic moment $\mu_s = 2.95 \mu_B$, respectively. The Hamiltonian $\mathcal{H}$ of $CrI_3$ monolayer reads $\mathcal{H} = \frac{S^2}{2}\sum_{\langle i,j \rangle}(J\mathbf{m}_i \cdot \mathbf{m}_j + \mathbf{D} \cdot \mathbf{m}_i \times \mathbf{m}_j) - \frac{KS^2}{2}\sum_i(\mathbf{m}_i \cdot \mathbf{z})^2 - \sum_i \mathbf{m}_i \cdot \mathbf{B}$, where $S = 3/2$ is the magnitude of spin, the $J$ is the Heisenberg exchange parameter, $\mathbf{D}$ is the Dzyaloshinskii-Moriya interaction (DMI) vector with both the in-plane component ($D_{xy}$) and out-plane component ($D_z$) to stabilize the skyrmion, $K$ is the anisotropy parameter and $\mathbf{B}$ is the external magnetic field, respectively. Noteworthily, the in-plane component of the DMI vector is perpendicular to the line between the two Cr atoms. In the case of perpendicular magnetic anisotropy (8 V), $J$ is set to −1.962 meV for ferromagnetic coupling, $D_{xy} = -0.189 J$, $D_z = -0.119 J$ and $K = 0.019 J$. As for the case of in-plane anisotropy (16 V), the $J$ is set to −1.992 meV, $D_{xy} = -0.210 J$, $D_z = -0.132 J$ and $K = -0.013 J$. We derived these parameters from previous studies[42,48] and, based on the skyrmion formation conditions in Supplementary Fig. 9c, selected 8 V and 16 V for atomic-scale spin dynamics simulations. Moreover, the thermal fluctuation is introduced with a stochastic field $\mathbf{H}_T = \boldsymbol{\eta}(step)\sqrt{2k_B T\alpha/(\gamma \mu_s \Delta t)}$, where $\boldsymbol{\eta}$(step) is a random vector from a standard normal distribution, $k_B$ is the Boltzmann constant, $T = 2$ K is the temperature, and $\Delta t = 10$ fs is the time interval. To reach equilibrium, we performed 1 ns of relaxation at each magnetic field point along the hysteresis loop.

## Data availability

The data supporting the findings of this study are available within the paper and its Supplementary Information, and have been deposited in the Figshare repository. Other data used in these experiments are available from the authors upon reasonable request. Source data are provided with this paper.

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

## Acknowledgments

Longjiang Deng (L.J.D.) and Bo Peng (B.P.) acknowledge support from the National Science Foundation of China (52021001). B.P. acknowledges support from the Sichuan Provincial Outstanding Youth Science Foundation Project (2025NSFJQ0018) and the "Hundred Talents Program" Cultivation Project of UESTC (A1098531023601549). H.P.L. acknowledges support from the National Science Foundation of China (51972046). L.J.D. acknowledges support from Sichuan Provincial Science and Technology Department (Grant No. 99203070). P. Y was supported by the National Key R&D Program under Contract No. 2022YFA1402802 and the National Natural Science Foundation of China (NSFC) (Grants No. 12374103 and No. 12074057).

## Author contributions

Bo Peng (B.P.) and Longjiang Deng (L.J.D.) conceived and designed the project, supervising all aspects. Y.L.W. prepared the samples and performed the RMCD measurements and Raman measurements assisted by B.P., and analyzed and interpreted the results assisted by C.D.Y., H.P.L., P.H.Z., J.L.X., D.F.L, L.B.Z., P.Y., H.Z.G., R.C.C., L.J.D., and B.P.. Z.Z.Z. and

P.Y. performed the theory calculations. Y.L.W. and B.P. wrote the paper with input from all authors. All authors discussed the results.

## Competing interests

The authors declare no competing interests.
