## [Transparent Peer Review file · Nature Communications]

Voltage-controlled topological spin textures in the monolayer limit

Corresponding Author: Professor Bo Peng

Version 1:

Reviewer comments:

Reviewer #1

(Remarks to the Author)

This manuscript reports the emergence of skyrmions when electrostatically dope a monolayer CrI₃. The main experimental signatures are the topological magneto-optical signals in RMCD, with supporting analysis through theoretical simulations. The results and claims to electrically manipulate the magnetic anisotropy and skyrmion states will attract a lot of interest in the field of spintronics. My main concern is that such topological RMCD signatures are missing in an earlier well-established work (<https://www.nature.com/articles/s41565-018-0135-x>) that also studies the monolayer CrI₃ within similar doping levels. I also have a number of technical questions that the authors will need to first address.

1. As noted above, the earlier Nature Nano paper reports the doping dependence of RMCD in a monolayer of CrI₃. How do the authors explain or understand this discrepancy? Is the reported topological RMCD signal reproducible in multiple monolayer devices?
2. The manuscript didn't show the estimated doping level using experimental data, even though it's mentioned that such values were extracted and compared to theory. The doping level needs to be explicitly stated along with the gate-dependent results.
3. Relevant to 2, the corresponding Fermi level shift should also be estimated using the doping level to compare with earlier theory work like ref 40.
4. Even if the authors claim that the magnetic anisotropy is completely in-plane with 16V backgate, the magnetization under an out-of-plane magnetic field should still undergo a polarizing transition to a finite RMCD signal, which is not the case in Fig. 1f. The lack of RMCD signal at higher fields is not consistent with the authors' current explanation, and it will need further discussion or measurements (e.g., under in-plane field).
5. The authors assign the spin textures to be type-I and type-II skyrmions. Their specific distinction is not clear to me, and the assignment doesn't seem very justified. Are they assigned due to the kinks in Fig. 2d and 2e at 14V? There is insufficient discussion on the physical significance or mechanism for the results in 2d&e.
6. Further data interpretation of spin textures based on their temperature dependences can be done to deepen the physical understanding.

Reviewer #2

(Remarks to the Author)

The current manuscript presents a study on the RMCD measurements on a gated monolayer CrI₃ device. The authors observed that as the voltage turns on, the saturated RMCD signal in the hysteresis loop decreases, and a peak emerges right at the switching of magnetization, resembling the topological Hall effect at the DC limit. The authors interpret this as evidence of the existence of the skyrmion, and the role of gating is to tune magnetic anisotropy and DMI continuously with electric field. The authors also did theoretical analysis to support this interpretation.

The paper is well written. Nevertheless, I have several questions before I can accept the topological Hall effect (THE) interpretation.

- The standard method to extract THE in a bulk sample is to compare the magnetization with anomalous Hall effect, then extract the difference between the two as the THE. In work, there is no direct measurement of magnetization.
- The direct measurement of magnetization of a 2D magnet is very difficult, but not impossible (for example: Nat Commun 13, 5369 (2022)). I am not suggesting the authors have to perform other kinds of measurements to measure magnetization, but

they should try very hard to rule out the other potential origins for this RMCD signal, before interpreting it as THE. For example, could the domain structure causes the peak? Did the authors do spatial imaging?

- The other thing that is a bit strange is that if the reduced saturated magnetization is due to the canting from out of plane to in-plane, the spin should cant back to c-axis when the field increases. I don't see that in the data.
- Finally, is it possible that the reduced RMCD signal is not due to the magnetization, rather because of electric field modulated the band structure, so that the RMCD spectrum changes?

Reviewer #3

(Remarks to the Author)

In this manuscript, the authors demonstrated a voltage-control of magneto-optical effect observed in monolayer CrI₃. By applying a voltage on the sample, a THE like RMCD loop has been observed, and the authors attributed the peak and valley on hysteresis loop as the feature of skyrmions. The results are interesting and provide a way to manipulate the magnetism of CrI₃. Before the manuscript be recommended for publication in Nature Communications, the following items should be addressed.

1. For the device used in the manuscript, a hBN with 9.8 nm has been used as a dielectric layer, when a 5V voltage is applied on the sample, there may exist a tunneling current. How to exclude the effect of electric current?
2. By applying different voltage, the authors state that there are two different kinds of skyrmions, what is the difference of these two types of skyrmions?
3. The authors demonstrate the evolution of magnetic structures, is it impossible to observe the magnetic structures (domains and skyrmions) directly?
4. The authors attribute the peaks on the RMCD loops as the generation of skyrmions, how the features of the eliminate processes?
5. For 2D materials, topological effect have also been observed in CrVI₆ (Nature Physics, <https://doi.org/10.1038/s41567-024-02465-5>) and Fe₃GeTe₂ (ACS Nano, <https://doi.org/10.1021/acsnano.3c11480>), the authors are encouraged to include a brief review of topological magneto-optical effects of 2D materials part.

Version 2:

Reviewer comments:

Reviewer #1

(Remarks to the Author)

The authors made efforts to address my questions. However, they failed to address my main concerns in Question 1, and my key technical Question 4(that underlies the credibility of their data interpretation). Also, their estimation of the doping level in response to Question 2 also seems questionable. Based on the lack of reliable responses, I cannot recommend its publication.

In the response of Question 1, the authors argued that the data in the mentioned Nature Nano paper does not break inversion symmetry (only doping, no electric field) and therefore differ from their data. However, the corresponding E-field measurements were presented in a Nature Materials paper (<https://www.nature.com/articles/s41563-018-0040-6>) for a monolayer, which still lacks the "peaks" discussed in this manuscript, and therefore, there is still a discrepancy.

In Question 4, I don't see any saturation of magnetization, in the figure that the authors indicate there should be. I don't think the authors provide a proper and direct response to this question, and my original concerns remain

In question 2, the authors didn't provide the device parameters (dielectric thickness, etc.) and just directly provide the estimate of up to $3 \times 10^{12}/\text{cm}^2$ with electrostatic doping. I don't think such high doping level is realistic because the dielectric breakdown limit will be first reached before such high doping. It indicates the authors lack proper estimation of their device structure that undermines the credibility.

Reviewer #2

(Remarks to the Author)

The authors have addressed most of my concern. I still disagree with the statement that RMCD can be used as a representation of magnetization. I understand that it has been commonly used in the community. However, in the context of this work, the main result is that the RMCD is not proportional to magnetization (due to topological Hall effect.) Furthermore, as the author discussed, the magnitude of RMCD depends not only on the magnetization, but also on the electronic structures. Therefore, I suggest the authors refrain from making such a statement, as it is logically inconsistent with the observation of topological Hall effect. The comparison with theory modelling is reasonable and sufficient.

Reviewer #3

(Remarks to the Author)

The authors have addressed all my concerns, I recommend the manuscript to be published as it is.

Version 3:

Reviewer comments:

Reviewer #1

(Remarks to the Author)

The authors this time are able to address my concerns and clarify that they will need a combined doping and E-field dependence to stabilize the skyrmion states, which was not clear to me at least in their earlier draft and reply. Just for the author's information, the mentioned E-field dependence of bilayer CrI₃ paper also contains E-field dependence of a monolayer device in their supplementary information, instead of only focusing on the bilayer scenario.

As for my second question, I made a typo by typing 3×10^{13} as 3×10^{12} . However, the authors indeed now give the specific device geometry information in their updated manuscript. While the achieved doping is large, it's still within the reported maximum dielectric breakdown voltage of hBN. I therefore have no issue with their reported numbers.

I can now recommend its publication.

RESPONSE TO REVIEWERS' COMMENTS

Reviewer #1 (Remarks to the Author):

This manuscript reports the emergence of skyrmions when electrostatically dope a monolayer CrI₃. The main experimental signatures are the topological magneto-optical signals in RMCD, with supporting analysis through theoretical simulations. The results and claims to electrically manipulate the magnetic anisotropy and skyrmion states will attract a lot of interest in the field of spintronics. My main concern is that such topological RMCD signatures are missing in an earlier well-established work (<https://www.nature.com/articles/s41565-018-0135-x>) that also studies the monolayer CrI₃ within similar doping levels. I also have a number of technical questions that the authors will need to first address.

Reply: We appreciate the reviewer's time in assessing our work and their acknowledgment of its potential to attract broad interest within the spintronics community. In response, we have elaborated on the uniqueness of this manuscript by citing the works recommended by the reviewers. Furthermore, we have fabricated a new monolayer CrI₃ device as advised, which confirmed the reproducibility of the experimental results, and conducted additional theoretical calculations to address the primary concerns raised. These insightful suggestions have significantly enhanced the quality of the manuscript.

1) As noted above, the earlier Nature Nano paper reports the doping dependence of RMCD in a monolayer of CrI₃. How do the authors explain or understand this discrepancy? Is the reported topological RMCD signal reproducible in multiple monolayer devices?

Reply: We thank the reviewer for their comment regarding why this work has been able to uniquely observe the topological RMCD signal. It is noteworthy that the devices used in the previous study published in *Nature Nanotechnology* employed a symmetric dual-gate architecture. Moreover, that investigation deliberately excluded electric field effects, focusing solely on doping effects in isolation. In contrast, the present study

utilizes an asymmetric single-gate device design and establishes that the emergence of the topological RMCD signal necessitates both doping-mediated modulation of magnetic anisotropy and electric field-induced breaking of inversion symmetry. Furthermore, the asymmetric device configuration inherently breaks inversion symmetry, which may promote the establishment of the Dzyaloshinskii–Moriya interaction (DMI). Taken together, these factors suggest that the absence of DMI—a fundamental prerequisite for the stabilization of topological spin textures—likely constitutes the principal reason why topological RMCD signals remained undetected in earlier works. More importantly, we fabricated a new device and also observed the topological RMCD signal under an appropriate voltage (Fig. R1). The experimental data for Device 2 has been added to the supplementary materials (Supplementary Fig. 6), with corresponding descriptions highlighted in red in the main text.

Fig. R1. **a**, Optical microscope image of CrI₃ nanosheets exfoliated from bulk CrI₃ and adhered to the PDMS surface, and the region enclosed by the red dashed line represents the atomic-thick CrI₃ area. **b**, False-color optical micrograph of the device 2, with the junction region represented by the area enclosed by a white dashed line. **c-g**, RMCD versus magnetic field at different gate voltages at 10 K. The black arrows indicate the direction of the field sweep.

2) The manuscript didn't show the estimated doping level using experimental data, even though it's mentioned that such values were extracted and compared to theory. The doping level needs to be explicitly stated along with the gate-dependent results.

Reply: We thank the reviewer for their valuable suggestions. We have now indicated

the doping density (top axes) in Fig. R2c and have replaced the corresponding Supplementary Fig. 9c in the supplementary materials.

Fig. R2. **a**, A schematic representation of the electric-field-induced breaking of spatial inversion symmetry in monolayer CrI₃, leading to the emergence of the DMI. **b**, A schematic illustration of the band-edge electronic structure in p-doped or n-doped CrI₃, where the direction of orbital angular momentum is indicated by color coding, and spin-up and spin-down states are denoted by arrows. **c**, The voltage (doping density) dependence of the factor η , the magnetic anisotropy energy (K), and the RCD_T intensity (at -0.04 T) is investigated. D is induced by the electric field, while K_{doping} is caused by electrostatic doping. Both contributions are derived from prior theoretical frameworks and quantified using interpolation methods. Notably, K_E shows negligible voltage dependence, indicating that the electric field has minimal effect on magnetic anisotropy.

3) Relevant to 2, the corresponding Fermi level shift should also be estimated using the doping level to compare with earlier theory work like ref 40.

Reply: We sincerely thank the reviewer for their helpful suggestions. According to ref 40, we estimated a Fermi level shift of approximately 0.33 eV at 16 V.

4) *Even if the authors claim that the magnetic anisotropy is completely in-plane with 16 V backgate, the magnetization under an out-of-plane magnetic field should still undergo a polarizing transition to a finite RMCD signal, which is not the case in Fig. 1f. The lack of RMCD signal at higher fields is not consistent with the authors' current explanation, and it will need further discussion or measurements (e.g., under in-plane field).*

Reply: We sincerely thank the reviewer for this highly professional comment, which prompted us to re-examine our data with greater care. To address the reviewer's observation, Fig. R3 presents an enlarged section of Fig. 1f. As the magnetic field increases, the RMCD signal exhibits a nearly linear rise with a very small slope, approaching a positive finite value. This trend is consistent with a weak out-of-plane magnetic susceptibility resulting from in-plane anisotropy. It is this weak out-of-plane magnetization that likely led to the impression that the RMCD signal approaches zero under stronger out-of-plane magnetic fields in Fig. 1f. Our RMCD results (Fig. 1c-f) are consistent with the previously reported voltage-induced weak anomalous Hall signal observed even under high magnetic fields (the anomalous Hall curve at 2.8 V in Fig. R4), a phenomenon attributed to the gate-voltage-driven transition of magnetic anisotropy from perpendicular to in-plane (*Nat Electron* **6**, 28–36, 2023)).

Fig. R3. A magnified view of Fig. 1f. The gray line represents the experimental data, while the solid red line corresponds to its linear fit.

[FIGURE REDACTED]

Fig. R4. Hysteresis loops of R_{AHE} at 2 K with varying V_{G} (*Nat Electron* **6**, 28–36, 2023).

5) The authors assign the spin textures to be type-I and type-II skyrmions. Their specific distinction is not clear to me, and the assignment doesn't seem very justified. Are they assigned due to the kinks in Fig. 2d and 2e at 14 V? There is insufficient discussion on the physical significance or mechanism for the results in 2d&e.

Reply: Indeed, the kinks observed at 14 V in Fig. 2d and 2e can also be used to distinguish between Type-I and Type-II skyrmions, as the voltage around 14 V marks the critical transition point between perpendicular magnetic anisotropy and in-plane magnetic anisotropy (Fig. R2c). To prevent readers from assuming that Type-I and Type-II skyrmions differ in topological nature, we have explicitly provided the following definition in the article: "Type-I skyrmions are topological spin structures stabilized by an out-of-plane magnetic field in systems with perpendicular magnetic anisotropy, whereas Type-II skyrmions are formed under in-plane magnetic anisotropy with the assistance of an out-of-plane field." This definition has also been included in the caption of Fig. 2. The main purpose of introducing this definition is to facilitate

understanding of how the evolution of magnetic anisotropy with voltage affects the formation of skyrmions under an out-of-plane magnetic field throughout the process. For example, the kinks in the topological peak magnetic field position shown in Fig. 2d is caused by the transition in magnetic anisotropy direction. This is because skyrmion formation primarily results from the competition among magnetic anisotropy, the Dzyaloshinskii–Moriya interaction (DMI), and the out-of-plane magnetic field. A qualitative change in magnetic anisotropy leads to an abrupt shift in the required out-of-plane magnetic field strength for skyrmion stabilization, accompanied by a sudden change in skyrmion density (Fig. 2e). The physical mechanisms underlying Fig. 2d and 2e have been addressed in the section detailing skyrmion formation mechanisms (Fig. R2). Moreover, theoretical simulations of the field-assisted skyrmion formation process under both anisotropy conditions confirm consistency with the experimental observations (Fig. 3 and Supplementary Fig. 10).

6) Further data interpretation of spin textures based on their temperature dependences can be done to deepen the physical understanding.

Reply: We thank the reviewer for their valuable comments, which are instrumental in improving this manuscript. Both temperature and out-of-plane magnetic field act as effective fields and play a crucial role in the formation and annihilation of topological spin textures. Fig. R5a shows the evolution of magnetic domains under a temperature of $T/J = 0.0044$ as the out-of-plane magnetic field is swept from $B/J = -0.214$ to $B/J = 0.214$. The results indicate that skyrmions form at an out-of-plane magnetic field of $B/J = 0.086$ and undergo complete annihilation at $B/J = 0.197$. When the temperature is increased to $T/J = 0.0879$, skyrmions also form at $B/J = 0.086$, but become fully annihilated at a lower magnetic field of $B/J = 0.171$ (Fig. R5b). The methodology employed in the atomic-scale spin simulations remains consistent with that used in the theoretical simulations at 16 V (Fig. 3 and methods), with the exception that temperature is treated as a variable. Fig. R5c and 5d present the phase diagrams of the topological charge density in monolayer CrI_3 as functions of temperature and magnetic field. The results indicate that the magnetic field required for the initial formation of

skyrmions remains unchanged with increasing temperature (highlighted by the black dashed line), whereas the magnetic field at which skyrmions undergo complete annihilation decreases as the temperature rises (highlighted by the green dashed line). This behavior is consistent with the variation of the topological RMCD signal with magnetic field and temperature observed experimentally (Fig. 4b and 4c). This consistency provides further support for the interpretation that the topological RMCD peaks originate from skyrmions. New theoretical simulation data along with the relevant text highlighted in red have been included in the Supplementary Note 2 and Supplementary Fig 12.

Fig. R5. a, A series of representative magnetic domain configurations at $T/J = 0.0044$ during a magnetic field (B/J) sweep from negative to positive maximum. A 100×100 supercell was used for the magnetic simulation of the two-dimensional spin lattice, with colors mapping the out-of-plane magnetic moment components (M_z). **b,** A series of representative magnetic domain configurations at $T/J = 0.0879$ during a magnetic field (B/J) sweep from negative to positive maximum. **c,** The topological charge (per 10,000 spin sites) as a function of temperature and magnetic field (sweeping from negative to positive). **d,** The time-reversal process corresponding to (c) (sweeping from positive to negative). The black arrows indicate the direction of the magnetic field sweep.

Reviewer #2 (Remarks to the Author):

The current manuscript presents a study on the RMCD measurements on a gated monolayer CrI₃ device. The authors observed that as the voltage turns on, the saturated RMCD signal in the hysteresis loop decreases, and a peak emerges right at the switching of magnetization, resembling the topological Hall effect at the DC limit. The authors interpret this as evidence of the existence of the skyrmion, and the role of gating is to tune magnetic anisotropy and DMI continuously with electric field. The authors also did theoretical analysis to support this interpretation.

The paper is well written. Nevertheless, I have several questions before I can accept the topological Hall effect (THE) interpretation.

Reply: We thank the reviewer for taking the time to assess our work and for recognizing its value. The reviewer's remarks and comments were very helpful in clarifying certain crucial aspects of our analysis, particularly regarding the topological Hall effect (THE) interpretation. The revised manuscript clarifies these points and includes new analysis. We hope the revisions and the response below fully address the reviewer's points.

1)The standard method to extract THE in a bulk sample is to compare the magnetization with anomalous Hall effect, then extract the difference between the two as the THE. In work, there is no direct measurement of magnetization.

Reply: We thank the reviewer for raising an important point regarding the interpretation of the magnetization measurement. Actually, conventional RMCD hysteresis loop measurements without topological peaks have been widely recognized as an effective method for magnetization characterization in the field of two-dimensional magnetic materials (purple shaded area in Fig. R1), similar to how the anomalous Hall effect is employed for magnetization measurement. Therefore, extracting the topological RMCD signal by subtracting the conventional RMCD background is analogous to isolating the topological Hall effect in bulk materials by removing the anomalous Hall component (Fig. R1). Moreover, this perspective is further supported by our theoretical simulations (Fig. 3). Furthermore, as you noted below, directly measuring the magnetic moment in two-dimensional magnets remains a significant challenge at liquid He

temperature — particularly for those with atomic-scale thickness. Even when such measurements are achieved, it is difficult to correlate them quantitatively with the RMCD signal, thereby providing little substantive support for the extraction of the topological RMCD component.

Fig. R1. RMCD versus magnetic field at different gate voltages at 10 K. The black arrows indicate the direction of the field sweep, while the purple arrows with spheres schematically represent the magnetic anisotropy. The purple shaded area corresponds to the conventional RMCD signal, which is proportional to the out-of-plane magnetization, while the orange peak represents the topological RMCD contribution.

2)The direct measurement of magnetization of a 2D magnet is very difficult, but not impossible (for example: Nat Commun 13, 5369 (2022)). I am not suggesting the authors have to perform other kinds of measurements to measure magnetization, but they should try very hard to rule out the other potential origins for this RMCD signal, before interpreting it as THE. For example, could the domain structure causes the peak? Did the authors do spatial imaging?

Reply: We thank the reviewer for their insightful comments on the correlation between topological RMCD signals and skyrmions. As you rightly noted, direct measurement of magnetic moments or magnetization in two-dimensional magnets remains highly challenging. While defect-mediated spin states allow detection in magnetic nanosheets—which may exhibit quasi-bulk behavior—direct probing in atomically thin layers is considerably more difficult.

Real-space observation of skyrmion remains exceptionally difficult for several reasons. First, the magnetic signal from a monolayer is extremely weak and requires

ultra-high-sensitivity detectors. Second, our calculations indicate a skyrmion diameter below 10 nm, posing severe spatial resolution challenges. Although scanning NV magnetometry offers high sensitivity, achieving ~ 10 nm resolution remains challenging; the current best spatial resolution for imaging magnetic domains in 2D materials is around 50 nm (*Science* **374**, 1140-1144, 2021). Third, such experiments require low-temperature conditions and in situ electric and magnetic field control. Additionally, CrI₃ is extremely air-sensitive and bare CrI₃ flakes drastically degrade within few seconds in ambient air, which exponentially increases the difficulty of the experiments.

A comprehensive experimental platform capable of real-space imaging of skyrmions in monolayer CrI₃ would need to simultaneously meet all these demanding criteria—a goal that entails methodological advances well beyond the scope of this study.

To further verify that the topological RMCD peaks indeed originate from skyrmions, we conducted additional theoretical simulations on the evolution of magnetic structures under various temperatures and magnetic fields at 16 V. Fig. R2a shows the evolution of magnetic domains under a temperature of $T/J = 0.0044$ as the out-of-plane magnetic field is swept from $B/J = -0.214$ to $B/J = 0.214$. The results indicate that skyrmions form at an out-of-plane magnetic field of $B/J = 0.086$ and undergo complete annihilation at $B/J = 0.197$. When the temperature is increased to $T/J = 0.0879$, skyrmions also form at $B/J = 0.086$, but become fully annihilated at a lower magnetic field of $B/J = 0.171$ (Fig. R2b). The methodology employed in the atomic-scale spin simulations remains consistent with that used in the theoretical simulations at 16 V (Fig. 3 and methods), with the exception that temperature is treated as a variable. Fig. R2c and 2d present the phase diagrams of the topological charge density in monolayer CrI₃ as functions of temperature and magnetic field. The results indicate that the magnetic field required for the initial formation of skyrmions remains unchanged with increasing temperature (highlighted by the black dashed line), whereas the magnetic field at which skyrmions undergo complete annihilation decreases as the temperature rises (highlighted by the green dashed line). This behavior is consistent with the variation of the topological RMCD signal with magnetic field and temperature observed experimentally (Fig. 4b and 4c). The dependence of the topological RMCD signal on temperature and magnetic

field closely mirrors the evolution of the topological charge density under identical conditions. This consistency provides further support for the interpretation that the topological RMCD peaks originate from skyrmions. New theoretical simulation data along with the relevant text highlighted in red have been included in the Supplementary Note 2 and Supplementary Fig 12.

Fig. R2. **a**, A series of representative magnetic domain configurations at $T/J = 0.0044$ during a magnetic field (B/J) sweep from negative to positive maximum. A 100×100 supercell was used for the magnetic simulation of the two-dimensional spin lattice, with colors mapping the out-of-plane magnetic moment components (M_z). **b**, A series of representative magnetic domain configurations at $T/J = 0.0879$ during a magnetic field (B/J) sweep from negative to positive maximum. **c**, The topological charge (per 10,000 spin sites) as a function of temperature and magnetic field (sweeping from negative to positive). **d**, The time-reversal process corresponding to (c) (sweeping from positive to negative). The black arrows indicate the direction of the magnetic field sweep.

It is worth emphasizing that in bulk magnets, skyrmions have been shown to generate topological magneto-optical Kerr peaks (Fig. R3b) analogous to topological RMCD signals (Fig. R3d and Fig. 4b), as both the magneto-optical Kerr effect and

RMCD share the same physical origin—the transverse optical Hall conductivity (*Phys. Rev. B* **75**, 214416, 2007; *Phys. Rev. Lett.* **108**, 087403, 2012; *Phys. Rev. B* **67**, 235203, 2003). Moreover, consistency has been demonstrated between the topological Hall effect in the DC limit and the topological magneto-optical peak in the same bulk magnet system (Fig. R3b and c).

Fig. R3. **a**, The AlB₂-type crystal structure of Gd_2PdSi_3 (left panel) and the schematic illustration of the triangular skyrmion lattice (SkL) (right panel). **b**, Magnetic phase diagram with magnetic field (H) parallel to the c axis and a contour map of topological Kerr rotation angle θ_K at 0.3 eV. IC-1 and IC-2 represent incommensurate spin-state phases, and PM represents the paramagnetic phase. The open circles represent phase boundaries determined by magnetization measurements. **c**, Magnetic-field dependence of the d.c. Hall conductivity σ_{xy} (left axis) and magnetization M (right axis) for $H \parallel c$ at 8 K. Hint represents the internal magnetic field, considering the demagnetization effect. The red shaded area denotes the SkL phase (*Nat. Commun.* **14**, 5416, 2023). **d**, The topological RMCD intensity as a function of temperature and magnetic field (sweeping from negative to positive).

3)The other thing that is a bit strange is that if the reduced saturated magnetization is due to the canting from out of plane to in-plane, the spin should cant back to c-axis when the field increases. I don't see that in the data.

Reply: We sincerely thank the reviewer for this highly professional comment, which prompted us to re-examine our data with greater care. To address the reviewer's observation, Fig. R4 presents an enlarged section of Fig. 1f. As the magnetic field increases, the RMCD signal exhibits a nearly linear rise with a very small slope, approaching a positive finite value. This trend is consistent with a weak out-of-plane magnetic susceptibility resulting from in-plane anisotropy. It is this weak out-of-plane magnetization that likely led to the impression that the RMCD signal approaches zero under stronger out-of-plane magnetic fields in Fig. 1f. Our RMCD results (Fig. 1c-f) are consistent with the previously reported voltage-induced weak anomalous Hall signal observed even under high magnetic fields (the anomalous Hall curve at 2.8 V in Fig. R5), a phenomenon attributed to the gate-voltage-driven transition of magnetic anisotropy from perpendicular to in-plane (*Nat Electron* **6**, 28–36, 2023).

Fig. R4. A magnified view of Fig. 1f. The gray line represents the experimental data, while the solid red line corresponds to its linear fit.

[FIGURE REDACTED]

Fig. R5. Hysteresis loops of R_{AHE} and magnetoresistance at 2 K with varying V_{G} (Nat Electron **6**, 28–36, 2023).

4) Finally, is it possible that the reduced RMCD signal is not due to the magnetization, rather because of electric field modulated the band structure, so that the RMCD spectrum changes?

Reply: We gratefully acknowledge the reviewer for their insightful comments. We have clarified that the electronic band structure, magnetization, and RMCD signal are fundamentally interrelated. Specifically, electric-field-induced doping modifies the band structure, which in turn alters the magnetization and correspondingly affects the RMCD signal.

The RMCD signal arises from the differential absorption of right- and left-circularly polarized (RCP and LCP) light within the material. RCP light carries an angular momentum of +1, while LCP light carries -1. Fig. R6 presents the electronic band structures of monolayer CrI_3 under both undoped and electron-doped conditions (*Nano Lett.* **20**, 929–935, 2019), with the total angular momentum (J) values indicated on the right.

In undoped (intrinsic) CrI₃, which exhibits perpendicular magnetic anisotropy (out-of-plane magnetization), RCP excitation induces electronic transitions that obey the optical selection rule requiring angular momentum conservation. Only the transition from $J = -3/2$ to $J = -1/2$ (green arrow in Fig. R6a) is allowed; all others are forbidden. In contrast, LCP excitation would require a decrease in angular momentum by 1, which violates conservation, resulting in negligible absorption. This strong contrast in absorption produces a large RMCD signal.

Under n-doping, the two lowest conduction bands—serving as the relevant unoccupied and occupied states—exhibit parallel spin alignment ($\sigma\sigma' = 1$, Fig. R6b). This alignment yields a negative magnetic anisotropy energy (MAE), stabilizing an in-plane magnetic anisotropy (in-plane magnetization). For a detailed explanation of how n-doping drives this band structure change and consequent magnetic reorientation, see Note R1. Concurrently, n-doping shifts the Fermi level near the conduction band minimum (Fig. R6b). The black fill indicates that the lower conduction band states become occupied. Under RCP excitation, Pauli blocking strongly suppresses the $J = -3/2 \rightarrow -1/2$ transition compared to the undoped case, markedly reducing RCP absorption. Thus, the n-doping-induced Fermi level shift not only switches the magnetic anisotropy (magnetization) from out-of-plane to in-plane but also strongly suppresses the RMCD signal. This confirms that the RMCD signal is intrinsically linked to both the magnetization and the underlying electronic structure.

Note R1: The magnetic anisotropy energy (MAE) in CrI₃ monolayers is intimately associated with the spin and orbital structures near the Fermi level. Under electron doping, the shift of the Fermi level toward the conduction band edge alters the magnetic anisotropy from the perpendicular type (perpendicular magnetization)—characteristic of the undoped case—to an in-plane anisotropy (in-plane magnetization). First-principles calculations have demonstrated that the MAE is primarily governed by the indirect SOC, with the direct SOC contribution being negligible (*2D Mater.* **4**, 035002, 2017). By the perturbation theory (*Phys. Rev. B.* **47**, 14932, 1993), the MAE can be succinctly expressed as

$$MAE = \xi^2 \sum_{\mu, o, \sigma, \sigma'} \sigma \sigma' \frac{|\langle o, \sigma | L_z^I | \mu, \sigma' \rangle|^2 - |\langle o, \sigma | L_x^I | \mu, \sigma' \rangle|^2}{E_{\mu, \sigma} - E_{o, \sigma'}}$$

Here, u and o denote the unoccupied and occupied states, respectively, $E_{u/o, \sigma}$ represents the band energy of these states, and the spin indices σ/σ' range over ± 1 , corresponding to the two orthogonal spin states at the k -point. Previous theoretical investigations have established that $|\langle o, \sigma | L_z^I | u, \sigma' \rangle|^2 - |\langle o, \sigma | L_x^I | u, \sigma' \rangle|^2$ is negative (*Phys. Rev. B.* **101**, 134418, 2020). In addition to the contribution from the orbital angular momentum, the relative spin polarization (σ, σ') of the states near the Fermi level must also be considered. When CrI₃ is n-doped, the two lowest conduction bands act as the unoccupied and occupied states, with their parallel spin alignment ($\sigma\sigma'=1$, Fig. R6b) resulting in a negative MAE, which favors an in-plane magnetic anisotropy (in-plane magnetization).

Fig. R6. a, b, A schematic illustration of the band-edge electronic structure in undoped or n-doped CrI₃, where the direction of orbital angular momentum is indicated by color coding, and spin-up and spin-down states are denoted by black arrows. The green arrow indicates the allowed electronic transition pathway under right-circularly polarized (RCP) light excitation in accordance with the selection rules, the purple arrow denotes the magnetic anisotropy or magnetization direction, and E_F represents the Fermi energy level.

Reviewer #3 (Remarks to the Author):

In this manuscript, the authors demonstrated a voltage-control of magneto-optical effect observed in monolayer CrI₃. By applying a voltage on the sample, a THE like

RMCD loop has been observed, and the authors attributed the peak and valley on hysteresis loop as the feature of skyrmions. The results are interesting and provide a way to manipulate the magnetism of CrI₃. Before the manuscript be recommended for publication in Nature Communications, the following items should be addressed.

Reply: We are greatly grateful to the reviewer for their interest in and recognition of the significance of our manuscript. We also appreciate the time and effort they have dedicated to evaluating our work. In response to their comments, we have performed additional experiments and theoretical simulations to further address the main concern raised. These insightful suggestions have significantly enhanced the quality of the manuscript.

1) For the device used in the manuscript, a hBN with 9.8 nm has been used as a dielectric layer, when a 5 V voltage is applied on the sample, there may exist a tunneling current. How to exclude the effect of electric current?

Reply: We thank the reviewer for their helpful comments. To characterize the tunneling current, the time-dependent variation of the current in the device was measured under a 8 V bias (Fig. R1). The tunneling current was found to be on the order of 10^{-8} mA, which is approximately eight orders of magnitude lower than the current levels previously shown to manipulate magnetic order (*ACS nano* **18**, 20055-20064, 2024). Thus, we conclude that any influence of the current on the magnetic order in our device can be ruled out.

Fig. R1. Leakage-time characteristics were measured at 10 K for the Gr/monolayer CrI₃/hBN/Gr devices.

2) By applying different voltage, the authors state that there are two different kinds of skyrmions, what is the difference of these two types of skyrmions?

Reply: We appreciate the reviewers' constructive feedback. The kinks observed at 14 V in Fig. 2d and 2e can be used to distinguish between Type-I and Type-II skyrmions, as the voltage around 14 V marks the critical transition point between perpendicular magnetic anisotropy and in-plane magnetic anisotropy (Fig. R2c). To prevent readers from assuming that Type-I and Type-II skyrmions differ in topological nature, we have explicitly provided the following definition in the article: "Type-I skyrmions are topological spin structures stabilized by an out-of-plane magnetic field in systems with perpendicular magnetic anisotropy, whereas Type-II skyrmions are formed under in-plane magnetic anisotropy with the assistance of an out-of-plane field." This definition has also been included in the caption of Fig. 2. The main purpose of introducing this definition is to facilitate understanding of how the evolution of magnetic anisotropy with voltage affects the formation of skyrmions under an out-of-plane magnetic field throughout the process. For example, the kinks in the topological peak magnetic field position shown in Fig. 2d is caused by the transition in magnetic anisotropy direction. This is because skyrmion formation primarily results from the competition among magnetic anisotropy, the Dzyaloshinskii–Moriya interaction (DMI), and the out-of-plane magnetic field. A qualitative change in magnetic anisotropy leads to an abrupt shift in the required out-of-plane magnetic field strength for skyrmion stabilization (Fig. 2d), accompanied by a sudden change in skyrmion density (Fig. 2e).

Fig. R2. **a**, A schematic representation of the electric-field-induced breaking of spatial inversion symmetry in monolayer CrI₃, leading to the emergence of the DMI. **b**, A schematic illustration of the band-edge electronic structure in p-doped or n-doped CrI₃, where the direction of orbital angular momentum is indicated by color coding, and spin-up and spin-down states are denoted by arrows. **c**, The voltage (doping density) dependence of the factor η , the magnetic anisotropy energy (K), and the RCD_T intensity (at -0.04 T) is investigated. D is induced by the electric field, while K_{doping} is caused by electrostatic doping. Both contributions are derived from prior theoretical frameworks and quantified using interpolation methods. Notably, K_E shows negligible voltage dependence, indicating that the electric field has minimal effect on magnetic anisotropy.

3) *The authors demonstrate the evolution of magnetic structures, is it impossible to observe the magnetic structures (domains and skyrmions) directly?*

Reply: We thank the reviewer for their helpful suggestions. Real-space observation of domain and skyrmion evolution remains extremely challenging with current

experimental technologies, and the air-sensitivity of CrI₃ flakes exponentially increases the experimental difficulties, as noted by Reviewer #2. Firstly, the magnetic signal from a monolayer is exceedingly weak, necessitating detectors with ultra-high sensitivity. Secondly, our theoretical calculations suggest that the skyrmion diameter is below 10 nm, rendering spatial resolution at this scale particularly difficult to achieve. Although scanning NV magnetometry measurements offer high sensitivity, achieving a spatial resolution of 10 nm remains extremely challenging. To the best of our knowledge, the highest spatial resolution currently achieved for observing magnetic domains in two-dimensional magnets is only about 50 nm (*Science* **374**,1140-1144, 2021). Thirdly, such measurements require a low-temperature environment coupled with the capability to apply in situ electric and magnetic fields. An experimental system capable of real-space imaging of skyrmion formation in monolayer CrI₃ must simultaneously satisfy all these demanding criteria—a task that will likely require the development of novel methodologies and extends far beyond the scope of the present study.

4) The authors attribute the peaks on the RMCD loops as the generation of skyrmions, how the features of the eliminate processes?

Reply: We thank the reviewer for their comments on the corresponding features of the skyrmion generation and annihilation processes reflected in the RMCD hysteresis loops. We have revealed that the emergence and disappearance of topological peaks correspond to the generation and annihilation of skyrmions through theoretical simulations of the out-of-plane magnetic field evolution of magnetic domains in monolayer CrI₃ under 8 V and 16 V. This observation is consistent with previously reported findings in non-centrosymmetric van der Waals bulk materials, where the emergence and disappearance of magneto-optical Kerr peaks were also associated with the generation and annihilation of skyrmions (*Nat. Phys.* **20**, 1145–1151, 2024).

It is worth emphasizing that in bulk magnets, skyrmions have been shown to generate topological magneto-optical Kerr peaks (Fig. R3b) analogous to topological RMCD signals (Fig. R3d and Fig. 4b), as both the magneto-optical Kerr effect and RMCD share the same physical origin—the transverse optical Hall conductivity (*Phys.*

Rev. B **75**, 214416, 2007; *Phys. Rev. Lett.* **108**, 087403, 2012; *Phys. Rev. B* **67**, 235203, 2003). Moreover, consistency has been demonstrated between the topological Hall effect in the DC limit and the topological magneto-optical peak in the same bulk magnet system (Fig. R3b and 3c).

Fig. R3. **a**, The AlB₂-type crystal structure of Gd_2PdSi_3 (left panel) and the schematic illustration of the triangular skyrmion lattice (SkL) (right panel). **b**, Magnetic phase diagram with magnetic field (H) parallel to the c axis and a contour map of topological Kerr rotation angle θ_K at 0.3 eV. IC-1 and IC-2 represent incommensurate spin-state phases, and PM represents the paramagnetic phase. The open circles represent phase boundaries determined by magnetization measurements. **c**, Magnetic-field dependence of the d.c. Hall conductivity σ_{xy} (left axis) and magnetization M (right axis) for $H \parallel c$ at 8 K. Hint represents the internal magnetic field, considering the demagnetization effect. The red shaded area denotes the SkL phase (*Nat. Commun.* **14**, 5416, 2023). **d**, The topological RMCD intensity as a function of temperature and magnetic field (sweeping from negative to positive).

To further verify that the topological RMCD peaks indeed originate from skyrmions, we conducted additional theoretical simulations on the evolution of

magnetic structures under various temperatures and magnetic fields at 16 V. Fig. R4a shows the evolution of magnetic domains under a temperature of $T/J = 0.0044$ as the out-of-plane magnetic field is swept from $B/J = -0.214$ to $B/J = 0.214$. The results indicate that skyrmions form at an out-of-plane magnetic field of $B/J = 0.086$ and undergo complete annihilation at $B/J = 0.197$. When the temperature is increased to $T/J = 0.0879$, skyrmions also form at $B/J = 0.086$, but become fully annihilated at a lower magnetic field of $B/J = 0.171$ (Fig. R4b). The methodology employed in the atomic-scale spin simulations remains consistent with that used in the theoretical simulations at 16 V (Fig. 3 and methods), with the exception that temperature is treated as a variable. Fig. R4c and R4d present the phase diagrams of the topological charge density in monolayer CrI₃ as functions of temperature and magnetic field. The results indicate that the magnetic field required for the initial formation of skyrmions remains largely unchanged with increasing temperature (highlighted by the black dashed line), whereas the magnetic field at which skyrmions undergo complete annihilation decreases as the temperature rises (highlighted by the green dashed line). This behavior is consistent with the variation of the topological RMCD signal with magnetic field and temperature observed experimentally (Fig. 4b and 4c). The dependence of the topological RMCD signal on temperature and magnetic field closely mirrors the evolution of the topological charge density under identical conditions. This consistency provides further support for the interpretation that the topological RMCD peaks originate from skyrmions. New theoretical simulation data along with the relevant text highlighted in red have been included in the Supplementary Note 2 and Supplementary Fig 12.

Fig. R4. **a**, A series of representative magnetic domain configurations at $T/J = 0.0044$ during a magnetic field (B/J) sweep from negative to positive maximum. A 100×100 supercell was used for the magnetic simulation of the two-dimensional spin lattice, with colors mapping the out-of-plane magnetic moment components (M_z). **b**, A series of representative magnetic domain configurations at $T/J = 0.0879$ during a magnetic field (B/J) sweep from negative to positive maximum. **c**, The topological charge (per 10,000 spin sites) as a function of temperature and magnetic field (sweeping from negative to positive). **d**, The time-reversal process corresponding to (c) (sweeping from positive to negative). The black arrows indicate the direction of the magnetic field sweep.

5) For 2D materials, topological effects have also been observed in CrVI_6 (Nature Physics, <https://doi.org/10.1038/s41567-024-02465-5>) and Fe_3GeTe_2 (ACS Nano, <https://doi.org/10.1021/acsnano.3c11480>), the authors are encouraged to include a brief review of topological magneto-optical effects of 2D materials part.

Reply: We sincerely thank the reviewer for their helpful suggestions. Indeed, topological magneto-optical effects have been previously observed in bulk van der Waals materials; however, voltage-controlled manipulation of skyrmions in the monolayer limit remains challenging and has not yet been reported. We have added relevant content to the introduction section, which is located at line nine.

RESPONSE TO REVIEWERS' COMMENTS

Reviewer #1 (Remarks to the Author):

The authors made efforts to address my questions. However, they failed to address my main concerns in Question 1, and my key technical Question 4 (that underlies the credibility of their data interpretation). Also, their estimation of the doping level in response to Question 2 also seems questionable. Based on the lack of reliable responses, I cannot recommend its publication.

In the response of Question 1, the authors argued that the data in the mentioned Nature Nano paper does not break inversion symmetry (only doping, no electric field) and therefore differ from their data. However, the corresponding E-field measurements were presented in a Nature Materials paper (<https://www.nature.com/articles/s41563-018-0040-6>) for a monolayer, which still lacks the "peaks" discussed in this manuscript, and therefore, there is still a discrepancy.

Reply: We thank the reviewer for their time and comments. We respectfully disagree with the overall assessment. We are concerned that there may be a misunderstanding of both the relevant literature and our work, as the critique seems to overlook the fundamental conclusions of the cited studies. We would be grateful if the reviewer could reconsider our points in light of a careful re-reading of the references.

The **Nature Nanotechnology paper** used dual-gated devices to **isolate the effect of doping** by canceling out the electric field. This work intentionally focused **solely on doping effects** while deliberately avoiding electric field effects (Fig. R1, highlighted by blue underlines). Due to the absence of an electric field to break inversion symmetry, the Dzyaloshinskii-Moriya interaction (DMI) is absent. Therefore, no skyrmion-related RMCD peak was observed.

Oppositely, the **Nature Materials paper**, also using dual-gated devices, specially excluded doping to **isolate the effect of the electric field**. By doing so, the study **focused solely on electric field effects** while avoiding doping effects (Fig. R2, highlighted by blue underlines). Due to the absence of doping to modulate magnetic anisotropy, no skyrmion-related RMCD peak is observed either, even though the electric field breaks inversion symmetry and generates DMI. As a supplementary note, it should be clarified that the Nature Materials paper on electric-field control **referenced by the reviewer actually investigated bilayer CrI₃, not the monolayer**. The monolayer (ferromagnetic) and bilayer (antiferromagnetic) systems are fundamentally different because of the interlayer antiferromagnetic coupling in the

bilayer. The distinct magnetic behaviors and switching mechanisms between bilayer and monolayer systems render such a comparison scientifically invalid. **To prevent any potential misunderstanding, we kindly request that the reviewer carefully re-examine the references they have cited.**

[FIGURE REDACTED]

Fig. R1. A partial screenshot of the Nature Nanotechnology article supplied by the reviewer (*Jiang, S. et al. Nature Nanotech 13, 549–553, 2018*).

[FIGURE REDACTED]

Fig. R2. A partial screenshot of the Nature Material article supplied by the reviewer (*Jiang, S. et al. Nature Mater 17, 406–410, 2018*).

It must be emphasized that skyrmions can only form when the magnetic anisotropy energy (MAE, K) is smaller than the DMI-dominated parameter η . As shown the phase picture in Fig. R3, it must be in the area of $K_{doping} < \eta$ highlighted by the red square. Therefore, the realization of skyrmions requires doping to modulate the MAE, while simultaneously applying an electric field to break inversion symmetry and induce the DMI. Only when the specific conditions of MAE and DMI— $K_{doping} < \eta$ —are met can skyrmions be stabilized. For the skyrmion-stabilization mechanism arising from the synergistic effect of electric field and doping, please refer to the relevant passages in the manuscript, which are marked in red.

Our work focuses not on electric field or doping effects alone, but on the regime where both coexist and act synergistically. This combined approach is essential for stabilizing skyrmions in our system.

Therefore, the absence of skyrmion-related features in the cited literature does not represent a contradiction, but rather confirms that neither electric field nor doping effects alone can easily generate skyrmions. Indeed, those studies were not designed to achieve the synergistic regime we have explored.

Fig. R3. Mechanism for generating magnetic skyrmions in monolayer CrI_3 by combined electric-field-induced inversion symmetry breaking and doping-controlled magnetic anisotropy (Please see the Supplementary Fig. 9).

In question 2, the authors didn't provide the device parameters (dielectric thickness, etc.) and just directly provide the estimate of up to $3 \cdot 10^{12} / \text{cm}^2$ with electrostatic doping. I don't think such high doping level is realistic because the dielectric breakdown limit will be first reached before such high doping. It indicates the authors lack proper

estimation of their device structure that undermines the credibility.

Reply: The reviewer questions the rationality of our estimated doping concentration ($\sim 3 \times 10^{12} \text{ cm}^{-2}$), suggesting it is unrealistic due to dielectric breakdown limitations. We believe this concern to lack factual basis and likely constitutes subjective speculation.

Our estimated maximum doping concentration in Fig. R3 (Supplementary Fig. 9c) is consistent with the experimental data reported in the Nature Nanotechnology reference cited by the reviewer (Fig. R4), where the use of hBN as the dielectric layer achieved a similar doping level of $\sim 2.8 \times 10^{13} \text{ cm}^{-2}$ in monolayer CrI₃. This value is approximately **an order of magnitude larger than the one stated by the reviewer.** Moreover, we disagree with the comment regarding the absence of our device parameters. In fact, the methodology section of our original manuscript explicitly states the specific device parameters used to estimate the doping concentration, with the relevant paragraph highlighted in red. We respectfully ask the reviewer to read carefully again.

[FIGURE REDACTED]

Fig. R4. The magnetic characteristics of monolayer CrI₃ as a function of gate voltage (bottom axis) and induced doping density (top axis) (*Fig. 2c, Jiang, S. et al. Controlling magnetism in 2D CrI₃ by electrostatic doping. Nature Nanotech 13, 549–553, 2018*).

In Question 4, I don't see any saturation of magnetization, in the figure that the authors indicate there should be. I don't think the authors provide a proper and direct response to this question, and my original concerns remain.

Reply: In systems with in-plane magnetic anisotropy, achieving out-of-plane magnetization and saturation generally requires very strong magnetic fields, as magnetizing along the hard axis is inherently challenging. As shown in Fig. R5, aside from the topological magneto-optical effect (the RMCD peak) originating from the skyrmion, the remainder of the RMCD signal—highlighted by the green shading and resulting from the out-of-plane magnetization of the magnetic background—gradually decreases with increasing voltage. This reveals that the magnetic anisotropy of CrI₃ shifts from out-of-plane to in-plane, which makes out-of-plane magnetization increasingly difficult. The voltage-induced reorientation of CrI₃'s magnetic anisotropy to the in-plane direction makes out-of-plane magnetization difficult (as indicated by the green-highlighted loop at 18 V). Therefore, the observed lack of saturation is not only a normal phenomenon but also strong evidence that the magnetic anisotropy has been switched from out-of-plane to in-plane.

Fig. R5. Voltage-controlled RMCD loop in other monolayer CrI₃ device (device 2). The hysteresis loop highlighted in green reveals that the magnetic anisotropy of CrI₃ is continuously tuned by voltage, gradually shifting from out-of-plane to in-plane.

To corroborate this point, we have added references from Nature Materials and Nature Electronics which show that under high fields, when voltage induces an out-of-plane to in-plane magnetic anisotropy transition, the out-of-plane magnetization remains difficult to saturate, with the signal intensity staying far below the 0 V baseline (Figs. R6, R7). These results are consistent with our reports that voltage induces an out-of-plane to in-plane magnetic anisotropy transition in monolayer CrI₃ (green-highlighted loop in Fig. R5) and provide direct support for our interpretation. A comprehensive analysis of our experimental results, together with prior findings, leads

us to conclude that the unsaturated phenomenon mentioned by the reviewers is not an incomprehensible or physically contradictory effect, but rather a normal manifestation of voltage-controlled magnetic anisotropy.

[FIGURE REDACTED]

Fig. R6. The out-of-plane hysteresis loops of the polar MOKE at varying gate voltages demonstrate that the voltage induces a transition of the magnetic anisotropy from out-of-plane to in-plane (*Tan, A.J. et al. Magneto-ionic control of magnetism using a solid-state proton pump. Nat. Mater. 18, 35–41 2019*).

[FIGURE REDACTED]

Fig. R7. Continuous manipulation of magnetic anisotropy in a van der Waals ferromagnet via electrical gating. R_{AHE} hysteresis loops at 2 K with varying V_G reveal that the applied voltage drives a transition from out-of-plane to in-plane magnetic anisotropy (*Tang, M. et al. Continuous manipulation of magnetic anisotropy in a van der Waals ferromagnet via electrical gating. Nat. Electron. 6, 28–36, 2023*).

Reviewer #2 (Remarks to the Author):

The authors have addressed most of my concern. I still disagree with the statement that RMCD can be used as a representation of magnetization. I understand that it has been commonly used in the community. However, in the context of this work, the main result is that the RMCD is not proportional to magnetization (due to topological Hall effect.) Furthermore, as the author discussed, the magnitude of RMCD depends not only on the magnetization, but also on the electronic structures. Therefore, I suggest the authors refrain from making such a statement, as it is logically inconsistent with the observation of topological Hall effect. The comparison with theory modelling is reasonable and sufficient.

Reply: We sincerely appreciate the valuable comments. From your comments, we feel that the reviewer is rigorous in academic research. We fully acknowledge that describing RMCD as strictly proportional to magnetization is not entirely accurate. Although RMCD primarily originates from magnetization in most collinear magnets, it can also generate RMCD signals in systems with topological magnetic structures or certain special magnetic configurations even in the absence of net magnetization. Accordingly, we have revised the relevant statements in the updated manuscript. To provide necessary clarification, in the manuscript we have decomposed the total RMCD in monolayer CrI₃ into two components: RCD_M and RCD_T , as defined in Equations (1)-(3) and the corresponding text. RCD_M originates from the magnetization of the ferromagnetic background, while RCD_T arises from the emergent magnetic field generated by the topological magnetic structure. Therefore, we understand that describing the total RMCD as proportional to magnetization is imprecise, particularly in systems possessing topological magnetic structures. Once again, we appreciate your insightful comments.

Reviewer #3 (Remarks to the Author):

The authors have addressed all my concerns, I recommend the manuscript to be published as it is.

Reply: We sincerely appreciate your support in the publication of our manuscript in Nature Communications.

RESPONSE TO REVIEWERS' COMMENTS

Reviewer #1 (Remarks to the Author):

The authors this time are able to address my concerns and clarify that they will need a combined doping and E-field dependence to stabilize the skyrmion states, which was not clear to me at least in their earlier draft and reply. Just for the author's information, the mentioned E-field dependence of bilayer CrI₃ paper also contains E-field dependence of a monolayer device in their supplementary information, instead of only focusing on the bilayer scenario.

As for my second question, I made a typo by typing 3×10^{13} as 3×10^{12} . However, the authors indeed now give the specific device geometry information in their updated manuscript. While the achieved doping is large, it's still within the reported maximum dielectric breakdown voltage of hBN. I therefore have no issue with their reported numbers.

I can now recommend its publication.

Reply: We sincerely appreciate your support in the publication of our manuscript in Nature Communications.